# Chloroplast Genomes of *Vitis flexuosa* and *Vitis amurensis*: Molecular Structure, Phylogenetic, and Comparative Analyses for Wild Plant Conservation

**DOI:** 10.3390/genes15060761

**Published:** 2024-06-10

**Authors:** Ji Eun Kim, Keyong Min Kim, Yang Su Kim, Gyu Young Chung, Sang Hoon Che, Chae Sun Na

**Affiliations:** 1Wild Plant Seed Office, Baekdudaegan National Arboretum, Bongwha 36209, Republic of Korea; jekim803@koagi.or.kr; 2Arboretum Education Office, Baekdudaegan National Arboretum, Bongwha 36209, Republic of Korea; 3Department of General Affairs, General Affairs Team, Gangeung-Wonju National University, Gangeung 25457, Republic of Korea; 4Department of Forest Science, Andong National University, Andong 36729, Republic of Korea; 5Forest Bioresources Department, Baekdudaegan National Arboretum, Bongwha 36209, Republic of Korea

**Keywords:** chloroplast, conservation, genetic diversity, genome, *Vitis*

## Abstract

The chloroplast genome plays a crucial role in elucidating genetic diversity and phylogenetic relationships. *Vitis vinifera* L. (grapevine) is an economically important species, prompting exploration of wild genetic resources to enhance stress resilience. We meticulously assembled the chloroplast genomes of two Korean *Vitis* L. species, *V. flexuosa* Thunb. and *V. amurensis* Rupr., contributing valuable data to the Korea Crop Wild Relatives inventory. Through exhaustive specimen collection spanning diverse ecological niches across South Korea, we ensured comprehensive representation of genetic diversity. Our analysis, which included rigorous codon usage bias assessment and repeat analysis, provides valuable insights into amino acid preferences and facilitates the identification of potential molecular markers. The assembled chloroplast genomes were subjected to meticulous annotation, revealing divergence hotspots enriched with nucleotide diversity, thereby presenting promising candidates for DNA barcodes. Additionally, phylogenetic analysis reaffirmed intra-genus relationships and identified related crops, shedding light on evolutionary patterns within the genus. Comparative examination with chloroplast genomes of other crops uncovered conserved sequences and variable regions, offering critical insights into genetic evolution and adaptation. Our study advances the understanding of chloroplast genomes, genetic diversity, and phylogenetic relationships within *Vitis* species, thereby laying a foundation for enhancing grapevine genetic diversity and resilience to environmental challenges.

## 1. Introduction

Food security has emerged as a global concern owing to climate change and population growth, and crop wild relatives (CWRs) are considered a principal solution to this problem. CWRs are wild plants closely related to crops and serve as potential wild ancestors or allied species. During domestication, crops undergo a substantial reduction in genetic variation, making them more susceptible to biotic and abiotic stresses and decreasing their capacity to withstand these stresses and reproduce [1]. In contrast, CWRs harbor a wealth of genetically valuable traits that allow them to adapt to diverse habitats and avoid genetic bottlenecks associated with domestication [2]. Therefore, CWRs are a valuable resource for increasing the genetic diversity of crops. This genetic improvement is essential for sustaining crop yields and food supply amidst ongoing and future climatic challenges.

Vitaceae is a family of flowering plants comprising ~950 species distributed across 16 genera [3]. Predominantly composed of climbing plants, this family is primarily found in the tropical regions of Asia, Africa, Australia, the Neotropics, and the Pacific Islands, with a few genera also present in temperate regions [3,4]. Among them, the genus *Vitis* L. is well known to include the most cultivated grape species, *V. vinifera* L. [5], and is the most economically important resource of fruits, wine, and raisins [6,7]. However, *V. vinifera* has historically faced challenges in terms of productivity owing to its susceptibility to pests, diseases, and abiotic stresses such as cold [5]. To address these limitations, the use of genes from wild plants has proven effective in enhancing biotic and abiotic tolerance and resistance [8,9,10]. Consequently, introducing wild genetic resources into breeding programs is crucial for the long-term sustainability and competitiveness of the grape industry in response to global environmental changes.

The chloroplast, a distinctive organelle in plant cells, plays a crucial role in essential metabolic pathways such as photosynthesis [11]. Generally, the size of the chloroplast genome ranges from 120 kb to 160 kb in angiosperms [12]. The chloroplast genome consists of four main structural components: a large single-copy (LSC, 80–90 kb) region, a small single-copy (SSC, 16–27 kb) region, and two inverted repeat (IR, 20–28 kb) regions [13,14]. Most angiosperm chloroplast genomes evolve slower than nuclear genomes and are primarily maternally inherited [15,16]. In contrast, chloroplast genomes in gymnosperms are typically paternally inherited [17]. Therefore, chloroplast genomes play a major role in classification, differentiation, and evolutionary processes. Recently, with advancements in sequencing technology and genetic information databases, there has been extensive research and analysis of chloroplast genomes of numerous plants [18,19,20,21]. Particularly, in recent years, the use of chloroplast genome sequences for species barcoding has emerged as a powerful approach in biological research. This approach offers several advantages, including rapid and cost-effective species identification, making it a valuable asset for various fields such as biodiversity conservation and agriculture [21,22]. This rapidly advancing field has potential for enhancing our understanding of plant biology and for devising effective strategies to conserve and sustainably manage plant biodiversity.

The aim of this study was to complete the chloroplast genomes of two *Vitis* species distributed in South Korea and contribute to the Korea Crop Wild Relatives (KCWRs) inventory. First, we assembled the chloroplast genomes, elucidated their basic structure, analyzed their genetic features, and explored the diversity of chloroplast genes through comparison among different crops. Subsequently, through phylogenetic analysis, we ascertained the overall trends within the genus and analyzed the phylogenetic relationships and positions relative to closely related crops. Overall, our results improve our knowledge of the chloroplast genome, genetic diversity, and phylogenetic relationships within *Vitis* species, thereby providing valuable perspectives for genetic diversity and wild plant conservation.

## 2. Material and Methods

### 2.1. Plant Materials

Six wild plants species belonging to the genus *Vitis* L. are distributed throughout South Korea. We collected the seeds of *V. flexuosa* Thunb. and *V. amurensis* Rupr. (Figure 1). These plants exhibit differences in terms of not only morphological features, such as fruit and leaf characteristics, but also their distribution in South Korea. The distribution analysis in the Republic of Korea revealed that *V. amurensis* is mainly distributed in the inland, whereas *V. flexuosa* is predominantly found in the southern and Jeju-do regions. Some of the seeds were stored in the Seed Bank at Baekdudaegan National Arboretum, Bongwha, South Korea, whereas others were grown into plants for use as materials. *Vitis flexuosa* samples were collected from Haan-dong, Gwangmyeong-si, Gyeonggi-do, Republic of Korea (37°27′33.1″ N, 126°52′9.4″ E) (seed bank accession number: B0021148). *Vitis amurensis* samples were collected from Hachu-ri, Inje-eup, Inje-gun, Gangwon-do, Republic of Korea (38°1′52.7″ N, 128°17′16.0″ E) (seed bank accession number: B002034).

### 2.2. DNA Extraction, Sequencing, Assembly, and Annotation

Total genomic DNA was extracted using the DNeasy Plant Mini Kit (Qiagen, Hilden, Germany). DNA quantity was measured using a NanoDrop 2000 (Thermo, Wilmington, DE, USA) and a Qubit 3.0 Fluorimeter (Invitrogen, Carlsbad, CA, USA). Next-generation paired-end sequencing was performed on a NovaSeq platform using TruSeq NanoDNA (Illumina, San Diego, CA, USA). We eliminated adapter sequences and low-quality reads (Phred score > 20), and then assembled the chloroplast genomes de novo using CLC Assembly Cell v4.2.1 (CLC Inc., Aarhus, Denmark). The genes were annotated using GeSeq [23] and Geneious Prime v2023.2.1 (Biomatters Ltd., Auckland, New Zealand) [24]. Finally, a circular map of the complete chloroplast genome was generated using CPGView [25]. The data supporting the findings of this study are available in GenBank, under accession numbers PP191159 and PP191162.

### 2.3. Analysis of Relative Synonymous Codon Usage Bias and Repeats

Relative synonymous codon usage (RSCU) was identified from protein-coding genes using the MEGA11 software [26]. According to Sharp and Li (1986), RSCU indicates the probability of selecting a particular codon among synonymous codons for the same amino acid [27]. An RSCU < 1 indicates less usage, RSCU = 1 suggests unbiased usage, and RSCU > 1 indicates high usage [28]. Additionally, RSCU values below 0.6 signify very low usage, whereas those exceeding 1.6 suggest very high usage [29,30,31]. Simple sequence repeats (SSRs) were identified using MISA [32], with mono-, di-, tri-, tetra-, penta-, and hexa-nucleotide sets of 10, 5, 4, 3, 3, and 3, respectively. A long repetitive sequence was identified, including forward (F), palindromic (P), reverse (R), and complement (C) repeats, with a set range from a minimum size of 20 bp to a maximum size of 50 bp, while setting a Hamming distance of 3 using REPuter [33].

### 2.4. Chloroplast Genome Structure and Comparative Analyses

We downloaded the chloroplast genomes of nine *Vitis* species from the NCBI GenBank (https://www.ncbi.nlm.nih.gov/, accessed on 23 November 2023) for these analyses. The chloroplast genome structure was determined using IRscope [34] to visualize and analyze the LSC, SSC, and IR region boundaries. The chloroplast sequences were aligned with those of *Ampelopsis japonica* (Thunb.) Makino, from the same subfamily (GenBank accession number: MK547541), using LAGAN alignment [35] within mVISTA [36,37].

### 2.5. Analysis of Divergence Hotspots

We used DnaSP v5 [38] to calculate nucleotide diversity (Pi) of the chloroplast sequences and identify the hotspots. Chloroplast genome sequences were aligned using MAFFT v7.490 [39] in Geneious Prime v2023.2.1 [24]. The step size for DnaSP v5 was set to the midpoint of 200 bp and a window size of 600 bp.

### 2.6. Phylogenetic Analysis

To explore phylogenetic relationships within the genus, we downloaded the chloroplast genomes from the NCBI GenBank. Additionally, the division of species characteristics was based on GRIN-Global [40], in which landraces refer to cultivated species. For phylogenetic analysis, protein-coding genes were selected using PhyloSuite v1.2.3 [41]. The sequences were concatenated into a single alignment and subsequently used to construct a phylogenetic tree employing the maximum likelihood (ML) method with IQ-TREE [42]. The best-fit model (K3Pu + F + I) was automatically selected, and 1000 ultrafast bootstrap replicates were employed to assess branch support.

## 3. Results

### 3.1. Features of the Complete Chloroplast Genomes

The complete chloroplast genomes of the two *Vitis* species exhibited considerable similarities in terms of gene composition, total length, and GC content. Specifically, *V. flexuosa* (GenBank: PP191159) and *V. amurensis* (GenBank: PP191162) have chloroplast genomes of 160,986 and 161,013 bp, respectively (Figure 2). The chloroplast genome has a quaternary structure consisting of LSC regions of 89,206–89,238 bp, IR regions of 26,360–26,354 bp, and SSC regions of 19,060–19,067 bp. Overall, the GC content of the two chloroplast genomes was 37.38% (Table 1).

In total, 133 genes were annotated, including 88 protein-coding, 37 tRNA, and 8 rRNA genes in each genome, of which 8 protein-coding (*ycf1*, *ycf2*, *ycf15*, *rps7*, *rps12*, *rpl2*, *rpl23*, and *ndhB*), 7 tRNA (*trnV^GAC^*, *trnR^ACG^*, *trnN^GUU^*, *trnL^CAA^*, *trnI^GAU^*, *trnI^CAU^*, and *trnA^UGC^*), and 4 rRNA (*rrn16*, *rrn23*, *rrn4.5*, and *rrn5*) genes were duplicated in the IR regions. Nine protein-coding (*rps16*, *atpF*, *rpoC1*, *petB*, *petD*, *rpl16*, *rpl22*, *ndhB*, and *ndhA*) and six tRNA (*trnK^UUU^*, *trnG^UCC^*, *trnL^UAA^*, *trnV^UAC^*, *trnI^GAU^*, and *trnA^UGC^*) genes contained one intron each, whereas three protein-coding genes (*rps12*, *ycf3*, and *clpP*) contained two introns each (Table 2). *rps12* is a trans-spliced gene with its 5′ exon located in the LSC region and 3′ exon located in the IR region. *matK* is located in the intron of *trnK^UUU^*.

### 3.2. Codon Usage Bias

RSCU was calculated from the protein-coding genes of the chloroplast genomes of each species. These genes contained 64 codons encoding 21 amino acids (Figure 3). The codon count ranged from 21,503 (*V. flexuosa*) to 21,541 (*V. amurensis*). Among the amino acids, leucine (Leu), encoded by UUA, UUG, CUU, CUC, or CUA, was the most abundant amino acid with 2258 codons (10.50–10.48%), whereas tryptophan (Trp), encoded by UGG, was the least abundant amino acid with 386 codons (1.80–1.79%). Although these species exhibit similar RSCU patterns, there are subtle differences among them in terms of the stop codons, particularly in the ratio of UAA (1.39 and 1.46%) to UGA (0.83% and 0.75%).

In both species, 30 codons showed RSCU values greater than 1, whereas 32 codons exhibited values less than 1. The UUA codon had the highest RSCU value (1.89), whereas the CGC codon had the lowest (0.34). Methionine (Met) and Trp showed unbiased codon use (RSCU = 1). Notably, codons ending in A or U bases (GUU, AGA, GGA, UUA, CCU, UCU, ACU, and UAU) showed very high biases (RSCU > 1.6), whereas 20 codons showed very low biases (RSCU < 0.6).

### 3.3. SSRs and Long-Repeat Sequences

In total, 86 SSRs were identified in the chloroplast genome of *V. flexuosa*, whereas 87 SSRs were identified in that of *V. amurensis*. These included 62 and 63 SSRs in the LSC regions of *V. flexuosa* and *V. amurensis*, respectively; 10 in the IR regions; and 14 in the SSC regions (Figure 4). Mono-nucleotides constituted the majority of the SSR motifs, accounting for 65.1–66.3% of the total repeats, ranging from 56 to 57. Additionally, 9 di-nucleotide (di-), 9 tri-nucleotide (tri-), 9 tetra-nucleotide (tetra-), and 9 penta-nucleotide repeat counts were recorded; however, no hexa-nucleotide motifs were identified. Among the identified repeat types, the mono-nucleotide repeat unit (A/T) was the most abundant, comprising 55–56 repeats.

In this study, 49 repeats were identified in each chloroplast genome and categorized into four types: 1 complementary repeat, 17 forward repeats, 23 palindromic repeats, and 8 reverse repeats (Figure 5). The sizes of the complex repeats varied from 20 bp to over 40 bp, with the majority falling within 30–39 bp.

### 3.4. Comparison of the Chloroplast Genome Sequences

We aimed to identify the characteristics of the chloroplast genomes shared by *Vitis* species. Eleven chloroplast genomes from nine crops were downloaded from GenBank. We compared the positions of the boundaries of the LSC/IRb/SSC/IRa regions and adjacent genes (Figure 6). The lengths of the IR regions in the genus ranged from 26,312 bp to 26,374 bp, showing similarities. Although changes in the length of the IR region were minimal, variations in the positions of boundary expansions and contractions were variable. *Vitis lavrusca* and *V. amurensis* (MN389556) had an LSC–IRb boundary (JLB) located outside *rps19*, whereas in the remaining species, it was positioned within the gene. In five species, the IRb–SSC boundary (JSB) was located 19 bp from *ndhF*, whereas in the other five species, it was located 40 bp from *ndhF*. Interestingly, in *V. flexuosa*, the IRb–SSC boundary was located at 38 bp from *ndhF*. Furthermore, the SSC–IRa boundary (JSA) was located within *ycf1*. In five species, it was positioned between the boundaries of 4456 bp and 1136 bp, whereas in the other five species, it was positioned between the boundaries of 4446 bp and 1116 bp. In *V. flexuosa*, it was located between the boundaries of 4559 bp and 1123 bp. The IRa-LSC boundary (JSB) was located 11 bp from *trnH* in almost all species, whereas in *V. labrusca*, it was positioned at a distance of 73 bp.

To analyze the variability in DNA sequences in these species compared with those in other crops, we compared 11 chloroplast genomes using mVISTA, with *A. japonica* as a reference (Figure 7). The coding regions exhibited sequences conserved among these species. In contrast, the non-coding regions showed high variability. Among these, high variability was identified in several regions of *trnQ^UUG^*-*rps16*, *psbC*-*rps14*, *trnT^UGU^*-*rps4*, and *ndhI*-*ndhG*.

### 3.5. Divergence Hotspots as Potential Markers

We used DnaSP v5 to analyze Pi (Figure 8). The Pi values of the 11 chloroplast genomes were in the range of 0–0.01145. *psbI*-*trnS^GCU^*-*trnG^UCC^*, trnS^UGA^-*psbZ*-*trnG^GCC^*-*trnfM^CAU^*, and *ycf*1-*ndhF* exhibited higher values (>0.006) than other regions. These are included in the LSC and IRb–SSC and distributed in non-coding gene regions, except for *ycf1*-*ndhF*.

### 3.6. Phylogenetic Relationships within Vitis Species

We constructed an ML phylogenetic tree consisting of 53 chloroplast genomes, which included 51 *Vitis* species (42 CWRs, 8 landraces, and 1 modern cultivar), along with two species of *Parthenocissus* Planch., which served as an outgroup (Figure 9). Based on our findings, we aimed to reconfirm the overall phylogenetic relationships within the genus, identify closely related crops with the two species, and integrate them into the KCWRs inventory.

The phylogenetic analysis revealed that the genus branches into three major clades, forming distinct clades within the genus, with some exceptions. This genus is divided into two subgenera, *Vitis* and *Muscadinia* (Planch.) Rehder, belonging to *Vitis* (subgenus). This clade was supported by bootstrap values of 72/77.4, 97/88, and 100/100. Both *V. flexuosa* and *V. amurensis*, whose chloroplast genomes were assembled in the present study, were included in clade 2, supported by a bootstrap value of 97/88.

## 4. Discussion

### 4.1. Chloroplast Genome Characteristics in the Two Vitis Species

The sizes of the complete chloroplast genomes of *V. flexuosa* and *V. amurensis* determined in this study (160,986 and 161,013 bp) were consistent with the genome size of angiosperms [12]. These genomes exhibit a quadruple structure, with LSC regions of 89,206 and 89,238 bp, IR regions of 26,360 and 26,354 bp, and SSC regions of 19,060 and 19,067 bp [13,14]. The observed similarity in genome sequence information within the genus suggests a shared genetic makeup among these species [18,19,20]. These results not only affirm the conservation of the chloroplast genome in *Vitis* but also support the maternal inheritance of chloroplasts, based on the length of the LSC, IR, and SSC regions [15,16].

The RSCU values indicate that most optimal synonymous codons end with A or U in higher plants [43]. The investigation of codon usage bias is essential for understanding the patterns of codon usage in closely related species [44,45]. This knowledge is valuable for exploring genetic evolution, deciphering gene expression characteristics, and providing guidance for breeding [46,47].

SSRs, also known as microsatellites, are repetitive DNA sequences that constitute a substantial proportion of higher eukaryotic genomes. These include one or a few bases that are repeated in tandem several times [48]. The distribution of SSRs in several chloroplast genomes is non-random and primarily characterized by mono-nucleotides, with A/T bases constituting the majority [49,50,51]. Long-repeat sequences are classified into four types with variations in size ranging from 20 to over 40 base pairs. This sequence information is used to identify phylogenetic relationships among species and serves as a molecular marker for analyzing genetic diversity [52,53,54].

### 4.2. Comparative Analysis of Chloroplast Genomes of Vitis Species with Those of Other Crops

Fluctuations in IR regions due to expansion and contraction are common occurrences in genome evolution. These variations contribute to shifts in the length of chloroplast genomes, and they can be useful for studying phylogenetic relationships and taxonomic classifications [55,56]. The IR region lengths ranged from 26,312 to 26,374 bp. Notably, several *Vitis* species exhibited variations in the boundary positions of the IRs, which were distinct from those of other species. Specific species such as *V. labrusca* and *V. flexuosa* were distinguished from other species in terms of their boundary positions. However, the overall changes in genome size were not significantly affected by the contraction and expansion of the IR regions. The mVISTA results, with *A. japonica* as the reference, showed conserved sequences within the coding regions; in contrast, the non-coding regions exhibited higher variability. This is common in the chloroplast genomes of most angiosperms [57,58] and genera [47]. DNA polymorphisms (Pi) ranged from 0 to 0.01145, and the nucleotide diversity was calculated for 11 chloroplast genomes. These nucleotide positions, primarily situated in the LSC and IRb–SSC regions, were predominantly composed of non-coding genes, with the exception of *ycf1-ndhF*. These non-coding genes exhibit higher variability and potentially greater discriminatory power, and when combined with coding gene markers, they could enhance identification accuracy and reliability. Identified as hotspot regions, these areas have the potential to serve as molecular markers for phylogenetic analyses and as DNA barcodes, thereby providing valuable genetic information for taxonomic studies.

### 4.3. Phylogenetic Relationships between Other Crops and CWRs of Vitis Species

We performed an ML phylogenetic analysis with 53 chloroplast genomes to reaffirm the overall phylogenetic relationships within the genus, identify crops closely related to the two species, and select them as CWRs. The phylogenetic analysis revealed that the genus branches into three major clades (North America, Eurasia, and Muscadinia) and is divided into two subgenera (*Vitis* and *Muscadinia*). The *Vitis* subgenus consists of groups from North America and Eurasia [3,59]. In a previous study, groups from Europe and Asia formed a sister group constituting the Eurasian clade. Moreover, they have been distinguished from the North American groups [60,61]. Furthermore, in this study, supporting inferences for this content were found at the IR boundary positions for each crop in each clade, with similar positions marked for the crops included in each clade. This clear partitioning of the main origins, maintaining geographical distribution, was further confirmed through phylogenetic analysis of the chloroplast genomes and genome structure in this study. In addition, some species distributed in North America identified within the Eurasian lineage may have undergone transitions, such as migration from North America to Eurasia through North Atlantic land bridges or long-distance dispersal [62]. *V. flexuosa* and *V. amurensis* formed clade 2 along with four crops, indicating a closer relationship with the four crops than with the others, with a supported bootstrap value of 97/88. In addition, we identified a high potential for the utilization of the two species as CWRs for these four crops (*V. vinifera*, *V. amurensis*, *V. coignetiae*, and *V. romanetii*). In the case of *V. amurensis* (PP191162 and MN389556), genetic distance was observed despite belonging to the same species. For example, PP191162 was collected in South Korea, whereas MN389556 was collected in China. These results suggest that geographical distance may contribute to genetic variation within species, highlighting the importance of considering geographic factors in genetic analyses. The major aims of this analysis were to elucidate the phylogenetic relationships within the genus, identify related crop species, and ultimately establish KCWRs. This study not only advances our understanding of the phylogenetic relationships within the genus but also elucidates the potential for utilizing these species as valuable genetic resources for crop improvement.

## 5. Conclusions

We comprehensively analyzed the chloroplast genome of *Vitis* species, confirming its genetic characteristics and demonstrating its potential as a CWR for crop improvement. The chloroplast genomes of *V. flexuosa* and *V. amurensis* were determined to be 160,986 and 161,013 bp, respectively, similar to those of common angiosperms. Analysis of chloroplast genomes, including gene composition, codon usage bias, SSRs, and long repeats, provided valuable insights. These findings not only contribute to our understanding of the genetic features of these genomes but also offer the potential for developing molecular markers and DNA barcodes. The phylogenetic analysis revealed three distinct clades within the genus, supported by two subgenera, and highlighted the close relationship of the two species with specific crops. In summary, our study provides insights into enhancing and improving the genetic diversity of crops facing challenges due to climate change and population growth. Additionally, it confirms the potential of the two species analyzed to serve as KCWRs. Following this comprehensive analysis, research could be pursued in different directions to further promote the utilization of *Vitis* species as valuable genetic resources for crop improvement. For example, future studies should investigate the functional significance of specific genes identified in the chloroplast genomes of *V. flexuosa* and *V. amurensis*. The genetic information obtained can be utilized to develop breeding strategies aimed at enhancing genetic diversity.

## Figures and Tables

**Figure 1 genes-15-00761-f001:**
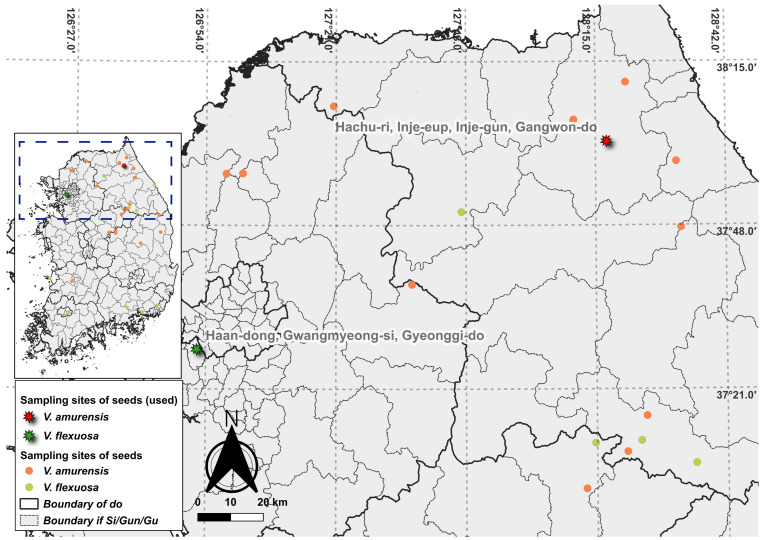
Seed sampling sites for *Vitis flexuosa* and *V. amurensis* in South Korea. The indicated colors and shapes identify seeds of each species that were used or unused.

**Figure 2 genes-15-00761-f002:**
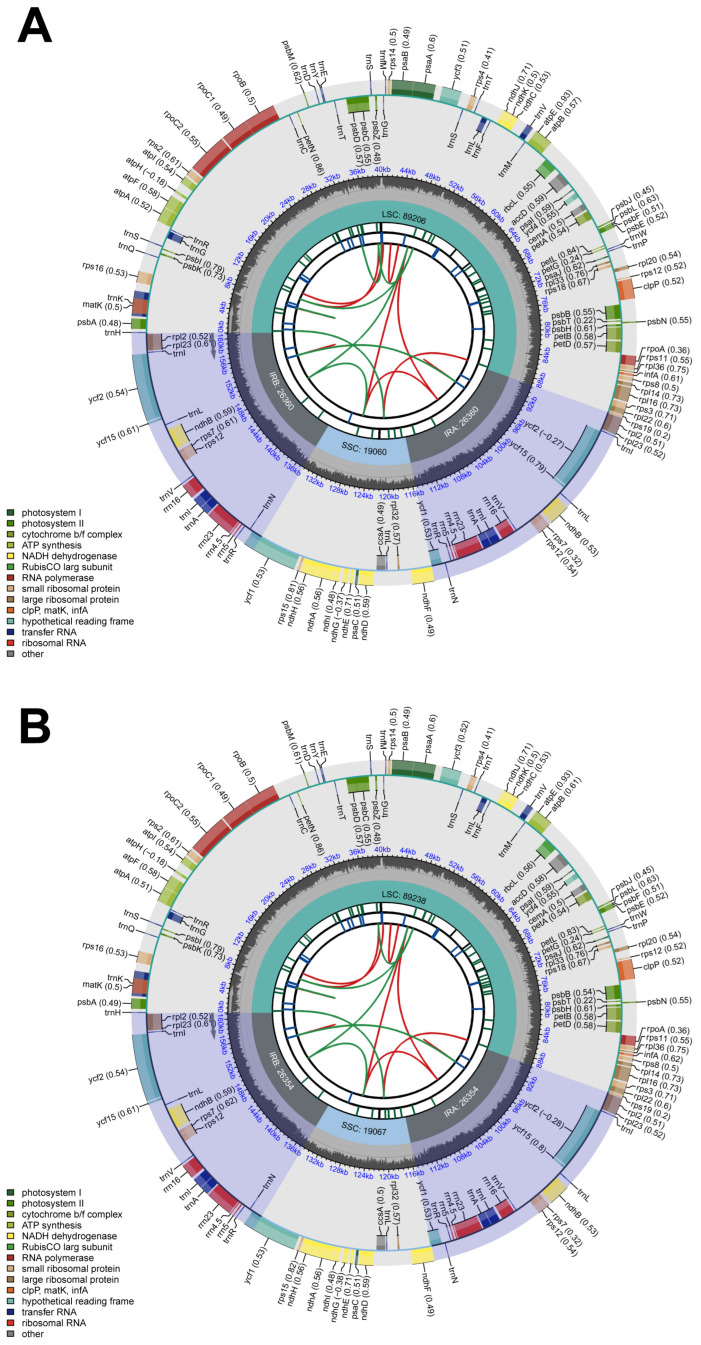
Circular maps representing the complete chloroplast genomes of (**A**) *V. flexuosa* and (**B**) *V. amurensis*. The center boundary shows the length of each region. Each gene has been color-coded to distinguish its functionality. Genes inside and outside the circle are transcribed in clockwise and counterclockwise directions, respectively, as indicated by the arrows.

**Figure 3 genes-15-00761-f003:**
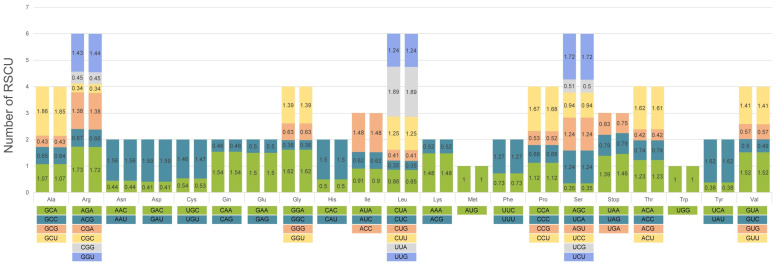
Relative synonymous codon usage for amino acids in the protein-coding regions of the chloroplast genomes of the two *Vitis* species. For each amino acid, the left bar represents *V. amurensis* and the right one represents *V. flexuosa*,.

**Figure 4 genes-15-00761-f004:**
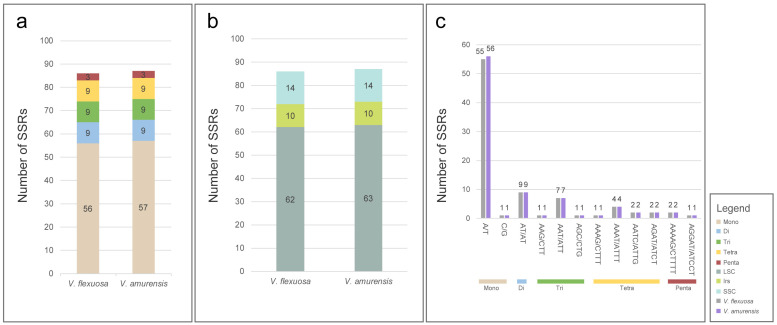
Simple sequence repeats (SSRs) in the chloroplast genomes. X-axes: species and types of SSRs. Y-axes: number of SSRs. (**a**) Number and types of SSRs. (**b**) Identification of SSRs in each region. (**c**) Frequency of SSRs in different repeat class types.

**Figure 5 genes-15-00761-f005:**
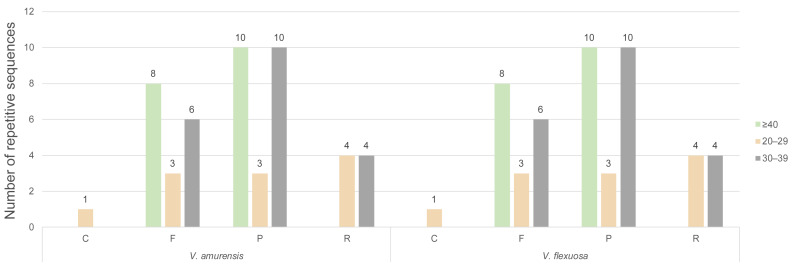
Number of long-repeat sequences in the *Vitis* chloroplast genomes (F: forward repeats; R: reverse repeats; P: palindromic repeats; C: complementary repeats).

**Figure 6 genes-15-00761-f006:**
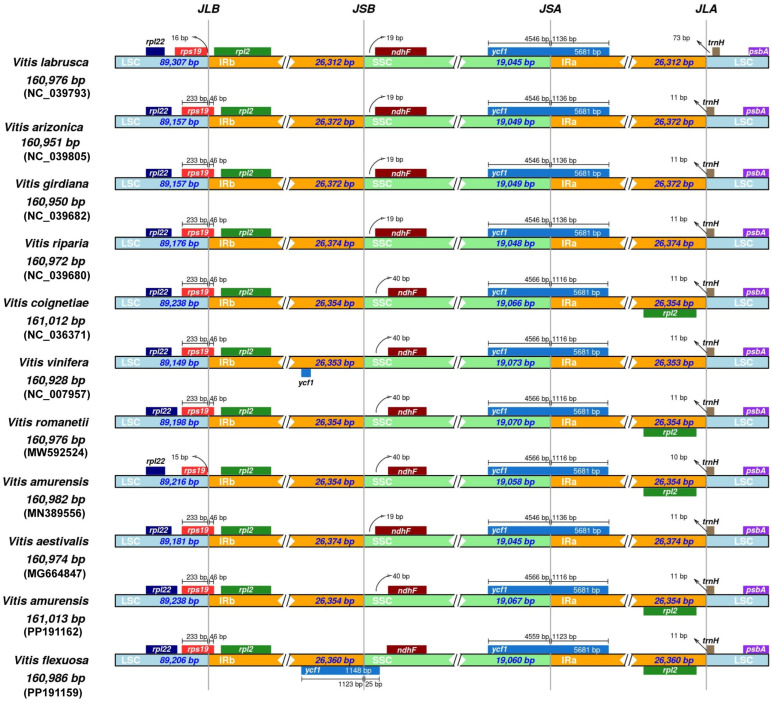
Comparison of boundary positions in LSC, SSC, and IR regions of *Vitis*. Genes are represented with boxes, and gaps between the genes and boundaries are indicated with the number of bases.

**Figure 7 genes-15-00761-f007:**
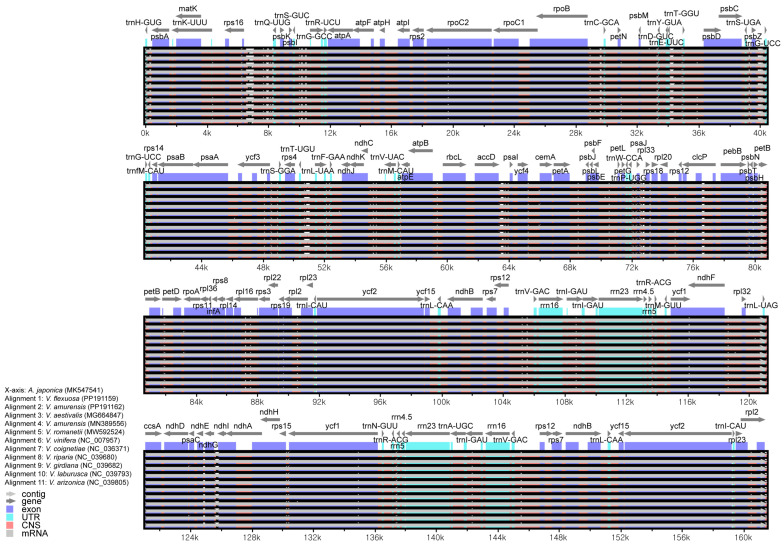
Comparison of three *Vitis* chloroplast genomes using the mVISTA program. Gray arrows indicate the direction and location of genes. Genomic regions, including non-coding (CNS) and coding regions, are depicted with red, light blue, and blue blocks.

**Figure 8 genes-15-00761-f008:**
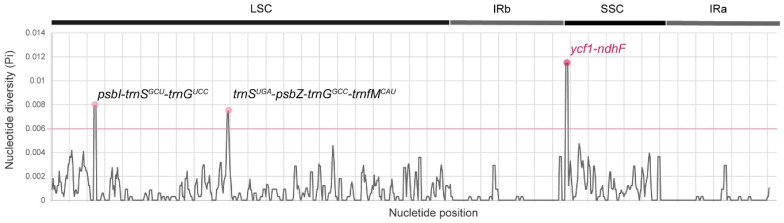
Sliding window analysis of chloroplast genomes of *Vitis*.

**Figure 9 genes-15-00761-f009:**
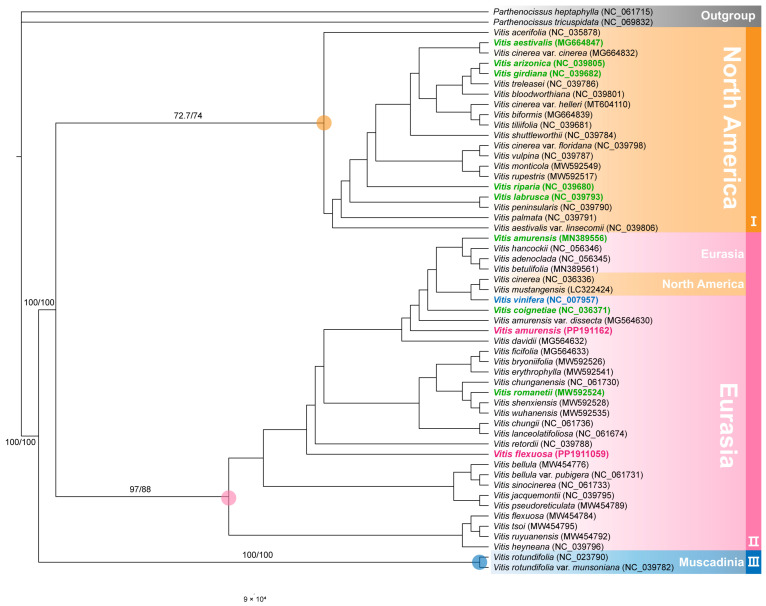
Phylogenetic tree constructed from *Vitis* chloroplast sequences, and bootstrap values based on 1000 replicates are displayed on each node. The tip label colors represent different categories: black for crop wild relatives (CWRs), green for landraces (native and also-cultivated), pink for subjects in this study, and blue for modern cultivars (criteria by GRIN-Global).

**Table 1 genes-15-00761-t001:** Summary of complete chloroplast genomes of *V. flexuosa* and *V. amurensis*.

Attribute	*V. flexuosa*	*V. amurensis*
GenBank no.	PP191159	PP191162
Total length (bp)	160,986	161,013
LSC length (bp)	89,206	89,238
IR length (bp)	26,360	26,354
SSC length (bp)	19,060	19,067
GC content (%)	37.38	37.38
Total genes	133 (114)	133 (114)
Protein-coding genes	88 (80)	88 (80)
tRNA genes	37 (30)	37 (30)
rRNA genes	8 (4)	8 (4)

The number in parentheses represents the number of unique genes, excluding duplicates.

**Table 2 genes-15-00761-t002:** List of annotated genes in the chloroplast genomes of *V. flexuosa* and *V. amurensis*.

Group of Genes	Gene Symbols
Photosystem I	*psaA*, *psaB*, *psaC*, *psaI*, *psaJ*, *ycf3* **, *ycf4*
Photosystem II	*psbA*, *psbB*, *psbC*, *psbD*, *psbE*, *psbF*, *psbH*, *psbI*, *psbJ*, *psbK*, *psbL*, *psbM*, *psbN*, *psbT*, *psbZ*
Cytochrome b/f complex	*petA*, *petB* *, *petD* *, *petG*, *petL*, *petN*
ATP synthase	*atpA*, *atpB*, *atpE*, *atpF* *, *atpH*, *atpI*
Rubisco	*rbcL*
NADH dehydrogenase	*ndhA* *, *ndhB*^2^ *, *ndhC*, *ndhD*, *ndhE*, *ndhF*, *ndhG*, *ndhH*, *ndhI*, *ndhJ*, *ndhK*
Proteins of large ribosomal subunits	*rpl14*, *rpl16* *, *rpl2*^2^ *, *rpl20*, *rpl22*, *rpl23*^2^, *rpl32*, *rpl33*, *rpl36*
Proteins of small ribosomal subunits	*rps11*, *rps12*^2^ **, *rps14*, *rps15*, *rps16* *, *rps18*, *rps19*, *rps2*, *rps3*, *rps4*, *rps7*^2^, *rps8*
RNA polymerase	*rpoA*, *rpoB*, *rpoC1* *, *rpoC2*
Acetyl-CoA carboxylase	*accD*
C-type cytochrome synthesis gene	*ccsA*
Envelope member protein	*cemA*
Protease	*clpP* **
Maturase	*matK*
Translational initiation factor	*infA*
Ribosomal RNAs	*rrn16*, *rrn23*, *rrn4.5*, *rrn5*
Transfer RNAs	*trnA^UGC^* *, *trnC^GCA^*, *trnD^GUC^*, *trnE^UUC^*, *trnF^GAA^*, *trnG^GCC^*, *trnG^UCC^* *, *trnH^GUG^*, *trnI^CAU^*, *trnI^GAU^* *, *trnK^UUU^* *, *trnL^CAA^*, *trnL^UAA^* *, *trnL^UAG^*, *trnM^CAU^*, *trnN^GUU^*, *trnP^UGG^*, *trnQ^UUG^*, *trnR^ACG^*, *trnR^UCU^*, *trnS^GCU^*, *trnS^GGA^*, *trnS^UGA^*, *trnT^GGU^*, *trnT^UGU^*, *trnV^GAC^*, *trnV^UAC^* *, *trnW^CCA^*, *trnY^GUA^*, *trnfM^CAU^*
Conserved ORFs	*ycf1*^2^, *ycf15*^2^, *ycf2*^2^

Gene^2^ = number of copies of a multi-copy gene; Gene * = genes containing a single intron; Gene ** = genes containing two introns; ORF = open reading frame.

## Data Availability

All data used are provided in the manuscript and are deposited in the NCBI GenBank (https://www.ncbi.nlm.nih.gov/, accessed on 23 November 2023) with accession numbers PP191159 and PP191162.

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
