# Peer review of "Chloroplast Genomes of Vitis flexuosa and Vitis amurensis: Molecular Structure, Phylogenetic, and Comparative Analyses for Wild Plant Conservation"

_genes, 2024, doi:10.3390/genes15060761_

Round 1

Reviewer 1 Report

Comments and Suggestions for Authors

The manuscript represents results of chloroplast genome analysis of two Vitis species. The authors performed study of the main chloroplast genome regions and their characteristics in these species. Despite interesting results there are some drawbacks.

Lines 62-63:"gymnosperms are typically paternally inherited". Were authors meant gymnosperms chroloroplast genome or anything else? Please, specify.

Introduction should contain several sentences about using of chloroplast genome for species barcoding.

Line 80: "Six wild plants belonging to the genus Vitis L. are distributed throughout Korea" Probably, it is about six wild plants species?

Figure 1: please, enlarge the figure to be more readable.

Figure 6: is it possible to represent these data in another format or represent the region of differences? The Figure is overwhelmed with data and it is dificult to differentiate blocks of various colours.

Line 254: "26,360–26,354 bp" Probably it is 26,354–26,360 bp. At the same time Lines 200-201 and Line 276 suggested the length of IR regions - 26,312 to 26,374 bp. Which data are correct?

Lines 256-258: "These results not only affirm the conservation of the chloroplast genome in Vitis but also support the maternal inheritance of chloroplasts" Please, provide additional explanation and confirmation of this suggestion. 

Line 284: Level of DNA polymorphism of 0-0.01145 is insufficient (very low). Why the authors suggest to serve these regions ""as molecular markers"(Line 288)? Please, provide explanation and additional information.

Since authors suppose using of non-coding sequences as molecular markers, in species barcoding, further information or comparison between coding markers (chloroplast genes that are used presently) and suggested by authors should be provided. Non-coding sequences are widely used for phylogenetic analysis or in intra-species variability research. What is the advantage of usinng them for species identification? And can the authors specify at least several relevant regions?

Thus, some additional information (discussion) could increase the significance of results obtained in research.

Author Response

첨부파일을 참조하시기 바랍니다.

Reviewer 2 Report

Comments and Suggestions for Authors

This study investigates the chloroplast genomes of two Vitis species and their potential as crop wild relatives (CWRs). The authors have conducted a comprehensive analysis, including genome assembly, annotation, comparative analysis, and phylogenetic analysis. This is a good paper! The spelling, grammar, and punctuation are of high standard, with no notable errors.

Comments and Suggestions for Authors: 

1. Why are some interesting and useful analyses (such as selection pressure and sequence divergence analyses) not done? By conducting these analyses and also performing comparative analyses with other species of Vitis (which are deposited in GenBank), useful results are obtained that will be very efficient for further studies.

2. The conclusions summarize the main findings and highlight the significance of the study. However, based on their findings, the authors could provide more specific recommendations for future research directions.

3. Formatting and Typos:

There are numerous minor formatting errors, including issues with spacing, and dashes. Please review the manuscript thoroughly to correct these errors.

4. The referenced figure requires additional explanation to be understandable. Consider revising both the figure and the accompanying text to enhance clarity.

5. Check and ensure consistency in citation style throughout the manuscript.

Round 2

Reviewer 1 Report

Comments and Suggestions for Authors

The authors made the most of the corrections. 

Lines 66-67: "In contrast, gymnosperms are typically paternally inherited". The question was to specify that it is chloroplast genomes are inhereted.

Figure 7 is still difficult to read and analyze but it is the authors' vision.

Lines 280-281: change "-" sign with "and" since the data for 2 species in the sentence "These genomes exhibit a quadruple structure, with LSC regions of 280 89,206–89,238 bp, IR regions of 26,360–26,354 bp, and SSC regions of 19,060–19,067 bp"

Lines 283-285: according to the authors' answer and manuscript the conclusion about maternal inheretance is made on the basis of the length of LSC , IR and SSC regions. That's why additional phrase would be appreciated.

Unfortunately, no additional related content can be found in the presented manuscript. Probably, it supposes the futher research of this issue.

Since authors suppose using of non-coding sequences as molecular markers, in species barcoding, further information or comparison between coding markers (chloroplast genes that are used presently) and suggested by authors should be provided. Non-coding sequences are widely used for phylogenetic analysis or in intra-species variability research. What is the advantage of usinng them for species identification? And can the authors specify at least several relevant regions?

Probably, it supposes the futher research of this issue. 

Thank you for the explanations and interesting research.
